# Engineering AQP1-Deficient DF-1 Suspension Cells for High-Yield IBDV Production and Vaccine Scale-Up

**DOI:** 10.3390/vaccines14010052

**Published:** 2025-12-31

**Authors:** Bingmei Dong, Ruonan Wang, Yu Guan, Xiubao Zhao, Ronghua Li, Qingqing Xu, Hui Li, Qingfang Gao, Shengjie Yao, Shuyu Song, Ashenafi Kiros Wubshet, Na Tang

**Affiliations:** 1College of Life Science, Cangzhou Normal University, Guofeng South Avenue 16#, Cangzhou 061001, China; dongbingmei_1998@126.com (B.D.); 13513171204@163.com (R.L.); 13091167583@163.com (S.Y.); 19831728876@163.com (S.S.); 2Shandong Binzhou Institute of Animal Husbandry and Veterinary Sciences, Binzhou 256600, China; q543685496@163.com (R.W.); 1589889981@163.com (Y.G.); xiari2006@126.com (X.Z.); zjzlxqq@126.com (Q.X.); yzlihui2020@163.com (H.L.); nafikw@gmail.com (A.K.W.); 3Shandong Lvdu Biological Technology Co., Ltd., Binzhou 256600, China; 18954339850@163.com; 4College of Veterinary and Animal Sciences, Mekelle University, Mekelle 2084, Ethiopia

**Keywords:** DF-1 cells, suspension-adapted culture, aquaporin-1 (AQP1), infectious bursal disease virus (IBDV)

## Abstract

**Background**: Large-scale production of poultry viral vaccines increasingly requires robust suspension cell platforms. However, most avian cell lines, including DF-1, are strictly anchorage-dependent, limiting scalability. Aquaporin-1 (AQP1) regulates cell–cell adhesion and membrane dynamics, making it a potential target for engineering suspension growth. This study aimed to generate a stable DF-1 suspension cell line via AQP1 disruption and evaluate its potential for enhanced infectious bursal disease virus (IBDV) production. **Methodology**: DF-1 cells were engineered using a CRISPR/Cas9 ribonucleoprotein system to create a truncated AQP1 gene. DF-1/AQP1^−^ cells were assessed for morphology, tumorigenicity in nude mice, and genetic stability across 20 passages. Suspension growth, cell density, and viability were measured. Cells were infected with IBDV strain BJQ902, and viral titers were compared with wild-type DF-1 and monolayer DF-1/AQP1^−^ cells. **Results**: DF-1/AQP1^−^ cells maintained normal morphology, were non-tumorigenic, and retained stable AQP1 mutations. They grew as true suspension cultures without adaptation, reaching 4.0 × 10^6^ cells/mL with >95% viability. Suspension DF-1/AQP1^−^ cells cells produced significantly higher viral titers (9.0 log TCID_50_/mL; 8.63 log EID_50_/mL) than both monolayer DF-1/AQP1^−^ and wild-type DF-1 cells. Virus production time was shortened in suspension cultures. **Conclusions**: Targeted AQP1 disruption converts DF-1 cells into a stable, non-tumorigenic suspension cell line with markedly enhanced IBDV production, providing a scalable platform for next-generation avian vaccine manufacturing.

## 1. Introduction

The infectious bursal disease virus (IBDV) is a double-stranded RNA virus that targets the avian immune system and causes an acute immunosuppressive disease in chickens, leading to substantial economic losses in the global poultry industry [1,2]. Currently, the prophylactic vaccinations of chicken flocks remain the primary strategy for controlling infectious bursal disease (IBD). Available vaccines mainly include live attenuated, inactivated, herpesvirus of turkeys (HVT) vector vaccines, and immunocomplex IBD vaccines. Among these, live and inactivated vaccines containing various classical or variant strains of IBDV are still the most widely used. However, the commercial production of IBDV live and inactivated vaccines relies heavily on specific-pathogen-free (SPF) embryonated chicken eggs [3,4]. This egg-dependent production system is constrained by limited egg availability, long cultivation cycles, and labor-intensive procedures [5]. These drawbacks have increasingly limited the scalability of traditional vaccine manufacturing and have driven the search for cost-effective alternatives and more robust cell-based production platforms.

Among the potential alternatives to traditional egg-based vaccine production, the use of primary cultured cells and immortalized cell lines has been extensively explored [6]. However, primary cell cultures remain constrained by the limited availability of chicken embryos or donor animals. Consequently, immortalized cell lines have gained increasing attention for their scalability and suitability for large-scale vaccine manufacturing [7,8]. Several cell-based viral vaccines, including poliovirus, rabies virus, and rotavirus vaccines produced in Vero cells, have already been successfully commercialized [9], and MDCK cells are now widely employed for influenza vaccine production [10,11]. In addition, other cell lines such as BHK-21, DF-1, and HEK293 have been used for virus propagation or vaccine development [12,13,14]. Despite their utility, most of these cell lines are anchorage-dependent and require surface attachment for proliferation [15,16]. Although they can yield consistently high viral titers and be expanded to industrial scales, large-scale cultivation typically relies on microcarrier beads that provide the necessary surface area in bioreactors [17,18]. This microcarrier-based strategy is considerably more cumbersome than cultivating cells directly in suspension [19,20]. Therefore, establishing a cell line capable of suspension growth would greatly streamline the scale-up process for viral vaccine production, simplify bioprocess monitoring, and provide improved control over contamination risks [21,22]. Currently, the swine foot-and-mouth disease vaccine has been successfully developed in BHK-21 cells, and the avian influenza vaccine has been developed in MDCK cells through full suspension cultivation using a bioreactor [5,6]. However, most viruses have not found the partner cells to facilitate vaccine production. Therefore, finding the partner cells and making them suitable for suspension culture is currently a pressing issue and especially for the traditional egg-based vaccine production.

Converting anchorage-dependent cells into anchorage-independent (suspension-adapted) cells would greatly simplify the production process and have strong potential to replace current microcarrier-based systems. Several strategies have been employed to adapt adherent cells to suspension growth, including the use of specialized low-serum or serum-free media in shaking or stirred systems, as well as genetic manipulation to modulate cell-adhesion pathways [23]. However, traditional adaptation methods are effective only for cell types that already possess some inherent potential for suspension growth, leaving most adherent cell lines difficult or impossible to adapt. As a result, genetic engineering approaches aimed at modifying anchorage dependence have been increasingly pursued. Previous studies have reported that genes such as human sialyltransferase and PTEN are closely associated with cellular adhesion processes [10,11,24]. Nonetheless, loss of adhesion alone does not guarantee successful suspension proliferation. Moreover, many adhesion-related genes, including PTEN, are tightly linked to oncogenic pathways, raising biosafety concerns for their application in vaccine production systems. Aquaporins (AQPs) are a family of membrane-channel proteins that facilitate the transport of water and small solutes across cellular membranes [25,26]. In recent years, AQPs have gained recognition for their broader biological roles, including contributions to cell adhesion, migration, and polarity, as well as their involvement in upstream regulation of numerous intracellular and intercellular signaling pathways that control cell proliferation, apoptosis, and invasiveness [27,28]. The AQP gene family is highly conserved across species, and 13 orthologs (AQP0–AQP12) have been identified in mammals and poultry [29,30,31,32].

Among these, AQP1 was first characterized in human erythrocytes and kidney cells by Peter Agre. AQP1 is a 28 kDa integral membrane protein containing six transmembrane domains arranged in two tandem repeats, two highly conserved Asn–Pro–Ala motifs, and cytoplasmic N- and C-terminal regions. Increasing evidence indicates that AQP1 plays an important role in regulating cell–cell adhesion and cell motility [33,34,35]. However, most previous studies have examined AQP1 in the context of human cancers or other pathophysiological conditions. Whether modulation of AQP1 can be leveraged to regulate cellular adhesiveness for the purpose of generating suspension-adapted cells suitable for large-scale vaccine production remains unclear. In this study, we successfully truncated the AQP1 gene in DF-1 cells using the CRISPR/Cas9 system, thereby reducing their dependence on surface anchorage. Furthermore, we demonstrate that the resulting suspension-adapted DF-1 cells support efficient propagation of IBDV and significantly enhance viral yield, highlighting their potential as an alternative platform for scalable vaccine production.

## 2. Materials and Methods

### 2.1. Cell Culture Systems and Virus Strains

DF-1 cells were obtained from the American Type Culture Collection (ATCC, Cat. No. CRL-12203, Rockefeller, MD, USA) and maintained in Dulbecco’s Modified Eagle Medium (DMEM; Sigma-Aldrich Corporation, St. Louis, MO, USA) supplemented with 10% (*v*/*v*) fetal bovine serum (FBS, Grand Island Biological Company, Great Island, NY, USA) at 37 °C in a humidified atmosphere containing 5% CO_2_. The IBDV strain BJQ902 was acquired from the Beijing Academy of Agriculture and Forestry Sciences (Beijing, China). The virus titer was determined to be 1 × 10^8.5^ TCID_50_/0.1 mL and 1 × 10^7.0^ EID_50_/0.2 mL.

### 2.2. Animal Models

Homozygous (nu/nu) adult nude mice, aged 5–8 weeks, were obtained from Charles River Laboratories (Beijing, China). Mice were housed in the Laboratory Animal Center of Shandong Lvdu Biological Technology Co., Ltd. (Binzhou, China), with ad libitum access to autoclaved water and gamma-irradiated food. Environmental conditions were maintained at 20–25 °C with a 12 h light/dark cycle. All experimental procedures were conducted in accordance with the Animal Handling and Research Ethics Guidelines of Shandong Lvdu Biological Technology Co., Ltd. At the end of the experiments, mice were humanely euthanized by manual cervical dislocation, followed by proper disposal via incineration under strict biosafety and biosecurity protocols. Ethical approval for all animal experiments was obtained from Shandong Lvdu Biological Technology Co., Ltd. (Reference Number: 202110001).

### 2.3. Generation of DF-1 Cells with AQP1 Deletion via CRISPR/Cas9

AQP1 is a 28 kDa membrane protein containing six transmembrane domains. To completely disrupt AQP1 function or interfere with its normal expression in DF-1 cells, the N-terminal transmembrane region of the AQP1 gene was selected as the target site. Several single guide RNAs (sgRNAs) targeting the N-terminal sequence (nucleotides 1–300 after the start codon) were designed using CRISPR guide sgRNA design tools “https://www.genscript.com/gencrispr-grna-design-tool.html” (accessed on 28 August 2021) and synthesized by Sangon Biotech (Shanghai, China).

Following construction of CRISPR/Cas9 plasmids and transfection into DF-1 cells, the editing efficiency of each sgRNA was assessed, and a pair of sgRNAs with high cleavage efficiency (target sequences listed in Table 1) was selected for subsequent experiments. In vitro transcription of the selected sgRNAs was performed using a one-step sgRNA transcription kit (Inovogen Tech. Co., Beijing, China). DF-1 cells were seeded in 24-well plates and grown to 70–90% confluence prior to transfection. The CRISPR ribonucleoprotein (RNP) complex was prepared by mixing 2.5 μL of in vitro–transcribed sgRNA with 2.5 μL of recombinant Cas9 protein and incubating for 10 min at room temperature. Next, 20 μL of CRISPR buffer was added to form the Cas9/sgRNA complex. Separately, the transfection reagent was prepared by combining 0.4 μL of Viromer with 24.6 μL of CRISPR buffer, and this mixture was rapidly combined with the Cas9/sgRNA complex in equal volumes. After incubation for 15 min at room temperature, the transfection mixture was added to the DF-1 cells. Seventy-two hours post-transfection, cells were subjected to single-cell cloning in 96-well plates using the limiting dilution method to isolate individual clones.

### 2.4. Identification of DF-1/AQP1^−^Cell Clones

#### 2.4.1. Identification of DF-1/AQP1^−^ Cell

Total RNA was extracted from each DF-1/AQP1^−^cell clone, parental DF-1 cells, DEPC-treated water, and complete cell culture medium and analyzed by one-step reverse transcription–polymerase chain reaction (RT-PCR) using the RT-PCR kit (AG11607, Aggurate Biology, Changsha, China) with primers listed in Table 1. The full-length AQP1 gene from each DF-1/AQP1^−^ clone was amplified and subjected to paired-end sequencing by Sangon Biotech (Shanghai, China). The resulting nucleotide and deduced amino acid sequences of the AQP1 mutants were compared with the wild-type AQP1 sequence to confirm gene editing. Additionally, three-dimensional structural models of the AQP1 protein in DF-1/AQP1^−^ and parental DF-1 cells were generated using SWISS-MODEL (https://swissmodel.expasy.org/ (accessed on 28 August 2021)), and the transmembrane domains were analyzed using TMHMM-2.0 (https://services.healthtech.dtu.dk/services/TMHMM-2.0/ (accessed on 28 August 2021)). Cells harboring the truncated AQP1 gene were further expanded and subjected to additional subcloning for downstream experiments.

#### 2.4.2. Determination of the Concentration of AQP1 Protein in DF-1/AQP1^−^ Cell with ELISA

DF-1 cells and DF-1/AQP1^−^ cells (1 × 10^6^ per well) were cultured in 6-well plates for 48 h at 37 °C with 5% CO_2_. Cells were harvested, subjected to three freeze–thaw cycles, and centrifuged at 1000× *g* for 20 min. The concentration of AQP1 protein in the resulting supernatants was quantified using a Chicken AQP1 ELISA kit (TW15375, Tongwei Bio, Chengdu, China). For the assay, microplate wells pre-coated with anti-chicken AQP1 capture antibodies received 50 µL of each standard (800, 400, 200, 100, 50, and 25 pg/mL) or cell sample in duplicate, followed by 100 µL of HRP-conjugated detection antibody. Plates were incubated for 60 min at 37 °C, washed four times, and then incubated with 50 µL each of Substrate A and Substrate B for 15 min at 37 °C. Reactions were stopped, and absorbance was measured at 450 nm using a microplate reader. A standard curve was generated from the mean OD values of the calibration standards, and AQP1 concentrations in the test samples were calculated using the fitted regression equation.

### 2.5. Morphological Observation of DF-1/AQP1^−^ Cells in T-Flasks

A total of 1 × 10^6^ DF-1/AQP1^−^ cells and wild-type DF-1 cells were continuously passaged for five generations in 25 cm^2^ T-flasks. During each passage, cells were detached either by gentle pipetting or by treatment with 0.25% trypsin-EDTA (Thermo Fisher, Waltham, MA, USA). The morphology of DF-1/AQP1^−^ and wild-type DF-1 cells was examined 48 h after each passage.

### 2.6. Growth Kinetics and Cell Viability of DF-1/AQP1^−^ Cells

#### 2.6.1. The Analysis About Growth Kinetics and Cell Viability of DF-1/AQP1^−^ Cells Monolayer Culture

A total of 1 × 10^6^ DF-1/AQP1^−^ cells and wild-type DF-1 cells were separately cultured in 25 cm^2^ T-flasks containing 10 mL DMEM supplemented with 10% (*v*/*v*) fetal bovine serum. Both cell types were continuously passaged for five generations. At 24 h, 48 h, and 72 h of each passage, cell numbers and the proportion of viable cells were determined using trypan blue exclusion.

#### 2.6.2. The Analysis About Growth Kinetics and Cell Viability of DF-1/AQP1^−^ Cells Suspension Culture

DF-1/AQP1^−^ cells and wild-type DF-1 cells were each seeded at 1.5 × 10^7^ cells in 125 mL vented shake flasks (Corning) containing 30 mL DMEM supplemented with 5% (*v*/*v*) fetal bovine serum. Cultures were maintained at 37 °C with 5% CO_2_ on an orbital shaker at 120 rpm and continuously passaged for five generations. Cell density and viability were assessed at 24 h, 48 h, 72 h, and 96 h after each passage.

### 2.7. Tumorigenicity Assay of DF-1/AQP1^−^ Cells

For tumorigenicity assessment, wild-type DF-1 cells, DF-1/AQP1^−^ cells, and BHK-21 cells were thoroughly washed with phosphate-buffered saline (PBS) to remove residual culture medium and subsequently resuspended in PBS at 5.0 × 10^7^ cells/mL. For each cell type, five adult nude mice were subcutaneously inoculated over the scapular region with 0.2 mL of the cell suspension. Tumor formation in the injected mice was monitored weekly.

### 2.8. Passage Stability of the AQP1 Mutant Gene in DF-1/AQP1^−^ Cells

To investigate whether the AQP1 mutant gene undergoes any revertant mutations during continuous culture, DF-1/AQP1^−^ cells were passaged 20 times in 125 mL vented shake flasks using the suspension culture method. The full-length AQP1 gene from the 0th, 10th, and 20th passage generations was amplified, and sequence homology was analyzed to assess potential genetic reversion.

### 2.9. Proliferation of IBDV Strain BJQ902 in DF-1/AQP1^−^ Cells

#### 2.9.1. Monolayer DF-1/AQP1^−^ Cells

According to the monolayer culture method, wild-type DF-1 cells or DF-1/AQP1^−^ cells were grown to confluence in 25 cm^2^ T-flasks. After removal of the culture medium, the virus was inoculated at a multiplicity of infection (MOI) of 0.005. Following a 1 h adsorption period at 37 °C, 10 mL of maintenance medium containing 2% fetal bovine serum was added to each flask. Infected cultures were incubated at 37 °C with 5% CO_2_, and samples were harvested once the cells exhibited approximately 90% cytopathic effect.

#### 2.9.2. Suspension DF-1/AQP1^−^ Cells

DF-1/AQP1^−^ cells were cultured in suspension at 37 °C with 5% CO_2_ and agitated at 120 rpm in 125 mL vented shake flasks. After 48 h of growth, the cells were infected with IBDV at a multiplicity of infection (MOI) of 0.005. Glucose concentrations in the culture medium were monitored every 12 h using a YSI 2700 Select biochemistry analyzer. When the glucose level decreased below 1 g/L, sterile glucose was supplemented to a final concentration of 4 g/L. After 36–48 h post-infection, the cultures were harvested, and virus titers were quantified by EID_50_ in embryonated chicken eggs and TCID_50_ in chicken embryo fibroblasts (CEF) using the Reed–Muench method.

### 2.10. Statistical Analysis

All experiments were independently performed at least three times, with a minimum of three samples per group in each experiment. Data was analyzed using IBM SPSS Statistics 20.0. Statistical significance was assessed by one-way ANOVA followed by the LSD multiple comparison test. Data are presented as mean ± standard deviation (SD), and differences were considered statistically significant at *p* < 0.05 or *p* < 0.01.

## 3. Results

### 3.1. Successful Construction of AQP1 Mutant DF-1 Cells (DF-1/AQP1^−^ Cells) by CRISPR/Cas9 System

To verify successful CRISPR/Cas9-mediated disruption of the AQP1 gene, the DF-1/AQP1^−^ cell clone was analyzed by RT-PCR and Sanger sequencing. RT-PCR analysis of DF-1/AQP1^−^ cells revealed the absence of an approximately 120 bp AQP1 fragment compared with wild-type DF-1 cells (Figure 1A). Sequencing confirmed a double-strand break at the AQP1-g1 sgRNA target site adjacent to the PAM sequence. At the cleavage site, a 123 bp deletion and a 3 bp insertion were observed relative to wild-type DF-1 cells (full-length AQP1: 813 bp) (Figure 1B).

Analysis of the predicted AQP1 protein sequence showed that the mutant gene encodes a 230-amino-acid protein. Between the first and second transmembrane domains, 40 amino acids were deleted, and two residues were altered, resulting in a shortened α-helix and modifications to the transmembrane domain structure (Figure 1C–E). Surprisingly, ELISA quantification indicated a higher AQP1 protein concentration in DF-1/AQP1^−^ cells (212 ± 3 pg/mL) compared with wild-type DF-1 cells (152 ± 5 pg/mL, *p* < 0.05). These findings confirm the successful generation of AQP1-mutant DF-1/AQP1^−^ cells.

### 3.2. Morphology of DF-1/AQP1^−^ Cells in Monolayer

Following AQP1 gene mutation, DF-1/AQP1^−^ cells exhibited morphology similar to wild-type DF-1 cells, displaying typical polygonal or fusiform shapes at 48 h post-passage (Figure 2A,B). DF-1/AQP1^−^ cells were more readily dispersed, whereas wild-type DF-1 cells remained more adherent. These findings indicate that loss of AQP1 does not affect cell morphology but reduces cellular adhesiveness.

### 3.3. Growth Kinetics and Cell Viability of DF-1/AQP1^−^ Cells

#### 3.3.1. Growth Kinetics and Cell Viability of DF-1/AQP1^−^ Cells Monolayer Culture

In monolayer culture, DF-1/AQP1^−^ cells and wild-type DF-1 cells exhibited comparable growth kinetics and viability (*p* > 0.05). Cell densities increased steadily in both lines, reaching peak levels of >2 × 10^6^ cells/mL at 72 h (Figure 2C). Viability remained above 95% from 24 to 48 h but declined to approximately 90% by 72 h (Figure 2D).

#### 3.3.2. Growth Kinetics and Cell Viability of DF-1/AQP1^−^ Cells Suspension Culture

The adhesiveness of DF-1/AQP1^−^ cells was markedly lower than that of wild-type DF-1 cells due to the deletion of the AQP1 gene. Therefore, the DF-1/AQP1^−^ cells were evenly dispersed in the medium, without cell clumps. Furthermore, the DF-1/AQP1^−^ cells exhibited stable growth over five consecutive passages in suspension, reaching a density of approximately 3.5 × 10^6^ cells/mL with 95–99% viability at 72 h. By 96 h, cell density increased to 4.0 × 10^6^ cells/mL, while viability declined to 88–90%.

In contrast, wild-type DF-1 cells reached a peak density of 1.7 × 10^6^ cells/mL with 92% viability after the first passage, and that was obviously lower than the DF-1/AQP1^−^ cells in suspension (*p* < 0.05). Subsequently, both the cell density and viability decreased significantly in subsequent passages (*p* < 0.05), and a large number of cells shrunk in size and were in a state of death. Furthermore, it is worth noting that many of the wild-type DF-1 cell clusters floated in the medium. By the third generation, the number and vitality of the wild-type DF-1 cells could no longer support the continued passage and cultivation.

These results indicate that DF-1/AQP1^−^ cells can be maintained in suspension culture without adaptation (Figure 3A,B).

### 3.4. Tumorigenicity Assay of DF-1/AQP1^−^ Cells

Mice injected with BHK-21 cells exhibited lethargy and reduced food intake starting at week 10, whereas those receiving DF-1 or DF-1/AQP1^−^ cells remained healthy throughout the 12-week period. At the end of the study, no tumors or nodules were observed in mice inoculated with DF-1 or DF-1/AQP1^−^ cells. In contrast, BHK-21 cells formed teratomas at the injection sites, with tumor diameters reaching approximately 10 mm (Figure 3C).

### 3.5. Stability of Mutant AQP1 in DF-1/AQP1^−^ Cells

During passage, the DF-1/AQP1^−^ cells did not display aging, and the cell viability was also suitable for continued passage culture up to the 20 th generation. Therefore, the AQP1 mutant gene in DF-1/AQP1^−^ cells remained stable over 20 consecutive passages. Sequence analysis of the 0 th, 10 th, and 20 th generations revealed 100% homology, with no evidence of revertant mutations. The gene sequence was identical to that of the original DF-1/AQP1^−^ clone, indicating that the AQP1 mutation is genetically stable.

### 3.6. Replication and Virus Titers Analysis of IBDV Strain BJQ902 in DF-1/AQP1^−^ Cells

IBDV strain BJQ902 replicated efficiently in both wild-type DF-1 cells and DF-1/AQP1^−^ cells under monolayer conditions, with virus titers measured at approximately 72 h post-infection. In suspension culture, DF-1/AQP1^−^ cells supported viral replication, with peak titers achieved at 36–48 h. The highest TCID50 and EID50 values are summarized in Table 2. No significant differences were observed between wild-type DF-1 and DF-1/AQP1^−^ cells in monolayer culture (*p* > 0.05). In contrast, DF-1/AQP1^−^ cells in suspension exhibited significantly higher TCID50 and EID50 values compared with monolayer culture (*p* < 0.01) (Table 2). All the results indicated that the mutant of the AQP1 gene merely altered the adhesiveness of DF-1 cells but did not affect the susceptibility of DF-1 cells to IBDV. And then, the suspension culture significantly increased the number of DF-1/AQP1^−^ cells per unit volume, thereby enhancing the viral titer.

## 4. Discussion

IBDV causes an acute and highly contagious disease, leading to substantial economic losses in the global poultry industry [1,2]. In China, chickens infected with IBDV are often treated with emergency antibody administration to reduce mortality. However, surviving birds remain immunosuppressed and become more susceptible to secondary infections, which can further compromise egg production in adult chickens [2,36]. Consequently, prophylactic vaccination of chicken flocks remains the most cost-effective, practical, and efficient strategy for controlling IBDV [37].

Currently, traditional vaccines remain the most widely used despite the development of numerous new IBDV vaccines. These conventional vaccines primarily include live IBDV vaccines, typically composed of a single IBDV strain, and inactivated vaccines, which may contain a single IBDV strain or combine IBDV with other viruses, such as Newcastle disease virus [38]. Regardless of the vaccine type, robust IBDV cultivation is essential for production. Many avian viruses, including IBDV, are currently propagated in embryonated eggs for vaccine production [39]. However, egg-based virus culture is labor-intensive and time-consuming. Replacing embryonated eggs with cell lines is therefore an attractive alternative. At present, the large-scale cell culture techniques widely used in production mainly include adherent culture, immobilized culture, and suspension culture. Among them, the main method of adherent culture is flask culture, which has many disadvantages, such as high labor intensity, low production efficiency, easy contamination, and uneven product quality. Cell fixation culture technology originated from immobilized enzyme technology, which mainly includes the microcarrier culture method, the hollow fiber culture method, and the microcapsule culture method. Among them, microcarrier culture technology is currently the most mature and widely applied technology, and the technology has no requirement for the capabilities of cell suspension culture. However, this technology has high requirements for the bioreactor and has many problems, such as high cost of microcarriers, difficulty in degradation and recycling of microcarriers, difficulty in replenishing materials during the production process, easy contamination etc. Large-scale suspension culture technology is like microbial fermentation technology. Cells are put into a bioreactor and kept in a dispersed suspension state along with the movement of the culture medium. When they grow to a relatively high density, they can be used for production. This method has the advantages of high cell density, a high degree of automation, easy operation and uniform product quality. Obviously, the Large-scale suspension culture technology has overcome some of the drawbacks of microcarrier culture technology. However, the Large-scale suspension culture technology requires cell lines that can adapt to suspension culture. Therefore, the domestication or modification of adherent cells has become a crucial step in achieving large-scale suspension culture. IBDV has been shown to infect a wide range of cell types, including chicken embryo fibroblasts (CEF), DF-1 cells, Vero cells, and rabbit kidney cell lines [40,41,42]. Nevertheless, the relatively low viral titers, the laborious preparation of cells, and the anchorage dependence of most cell lines remain significant barriers to large-scale vaccine production in bioreactors. It is not surprising that many cell lines, including Chinese Hamster Ovary (CHO) cells, BHK-21 cells, and MDCK cells, have been successfully adapted to suspension culture [43,44,45]. These suspension-adapted cell lines are widely used in the industrial production of vaccines and monoclonal antibodies [46]. Consequently, converting an anchorage-dependent cell line to a suspension culture system could greatly simplify IBDV vaccine production. In previous studies, DF-1 cells were shown to be more suitable for IBDV propagation compared with Vero and RK-13 cells, making them the optimal choice for virus production in the present study. The results demonstrate that CRISPR/Cas9-mediated mutation of the AQP1 gene significantly reduces the anchorage dependence of DF-1 cells which eliminates the cumbersome procedures of suspension culture training and effectively increases the cell culture density. Moreover, DF-1/AQP1^−^ cells are non-tumorigenic and maintain stable expression of the mutant AQP1 gene, making them well-suited for large-scale suspension culture.

AQP1 is a water-channel protein that plays a key role in regulating cell adhesiveness, migration, and proliferation [28]. Silencing of AQP1 has been shown to markedly reduce the expression of adhesion molecules, including VCAM-1 and ICAM-1 [34]. Moreover, a recent study reported that blockade of AQP1 in epithelioid malignant pleural mesothelioma cells significantly inhibited both proliferation and migration in vitro [35]. In this study, DF-1/AQP1^−^ cells with a truncated AQP1 gene were successfully generated using the CRISPR/Cas9 system. Consistent with previous reports, the observed modifications in the AQP1 gene and protein structure are likely associated with reduced cell–cell adhesiveness, enabling DF-1/AQP1^−^ cells to be directly cultured in suspension without domestication. However, in suspension, the density and the viability of wild-type DF-1 cells obviously decreased with the continued passage, and that could not support the fourth passage and cultivation. This might be due to the fact that the wild-type DF-1 is not suitable for direct suspension culture. As a result, the rotation and shaking of the medium caused cell death, or apoptosis. But it is unlike prior studies; notably, the growth kinetics and cell viability of DF-1/AQP1^−^ cells were not significantly diminished compared with wild-type DF-1 cells in monolayer, which may reflect differences in cell type or cellular state. Protein structure predictions further indicated that the truncation of the AQP1 gene resulted in only partial alterations to the overall protein structure. Therefore, we conclude that the water-channel function of the AQP1 protein in DF-1/AQP1^−^ cells may be impaired, even though the protein remains expressed. Interestingly, the AQP1 protein concentration in DF-1/AQP1^−^ cells were significantly higher than in wild-type DF-1 cells. Whether the partial truncation of the AQP1 gene contributes to this increased expression warrants further investigation.

While suspension cell lines can simplify vaccine production by reducing complex processing, expansion, and monitoring steps, an ideal production cell line must also support high viral yields. Currently, traditional IBDV vaccine production in China primarily relies on inoculating the allantoic cavity of 9- to 11-day-old specific-pathogen-free chicken embryos, followed by harvesting allantoic fluid, embryo body, and allantoic membrane. Typically, 10–15 mL of tissue fluid is obtained per embryo, yielding a viral titer of approximately 7.7 log EID_50_/0.1 mL, and the cost of each milliliter is approximately 0.4–0.6 Chinese yuan based on the current price of SPF chicken embryos. In the present study, at the same multiplicity of infection (MOI = 0.005), the IBDV titer in DF-1/AQP1^−^ suspension cultures was 8.63 logEID_50_/0.1 mL which is significantly higher than those in the embryo and the DF-1 cells or DF-1/AQP1^−^ cells cultured as monolayers. Furthermore, the price per milliliter of the virus solution is only 0.055 Chinese yuan based on the current price of medium. Obviously, the value of each milliliter of IBDV in suspension culture decreases by 7.28 times to 10.91 times. Moreover, the culture time for IBDV in suspension was significantly shorter than that in monolayer culture. This improvement is likely due to the higher cell density achievable in suspension and the more uniform distribution of virus throughout the medium facilitated by stirring or shaking. Consequently, culture duration is reduced, and viral production is markedly enhanced. In addition, IBDV titers in DF-1/AQP1^−^ suspension cultures were significantly higher than those obtained from chicken the embryos. Extrapolating this system to a 10 L suspension culture at 4 × 10^6^ cells/mL, the potential virus yield would be equivalent to that from approximately 5000 to 9000 embryonated eggs. Therefore, large-scale suspension culture of IBDV using DF-1/AQP1^−^ cells not only improves production efficiency but also substantially increases virus yield.

In conclusion, our study demonstrates that DF-1/AQP1^−^ cells are well-suited for suspension culture of IBDV. Future work will focus on further elucidating the structure and functional consequences of the truncated AQP1 gene and protein in DF-1/AQP1^−^ cells, as well as optimizing culture conditions to enable full-scale suspension production of IBDV in bioreactors.

## 5. Conclusions

Our study demonstrates that truncation of the AQP1 gene markedly reduces the adhesiveness of DF-1 cells. Although adhesion is not completely abolished, DF-1/AQP1^−^ cells can be readily maintained in suspension without prior adaptation. Notably, IBDV propagated in DF-1/AQP1^−^ suspension cultures achieved significantly higher viral titers compared with DF-1 monolayer cultures or embryonated egg systems. These findings provide a promising foundation for the development of IBDV vaccine production using cell culture platforms.

## Figures and Tables

**Figure 1 vaccines-14-00052-f001:**
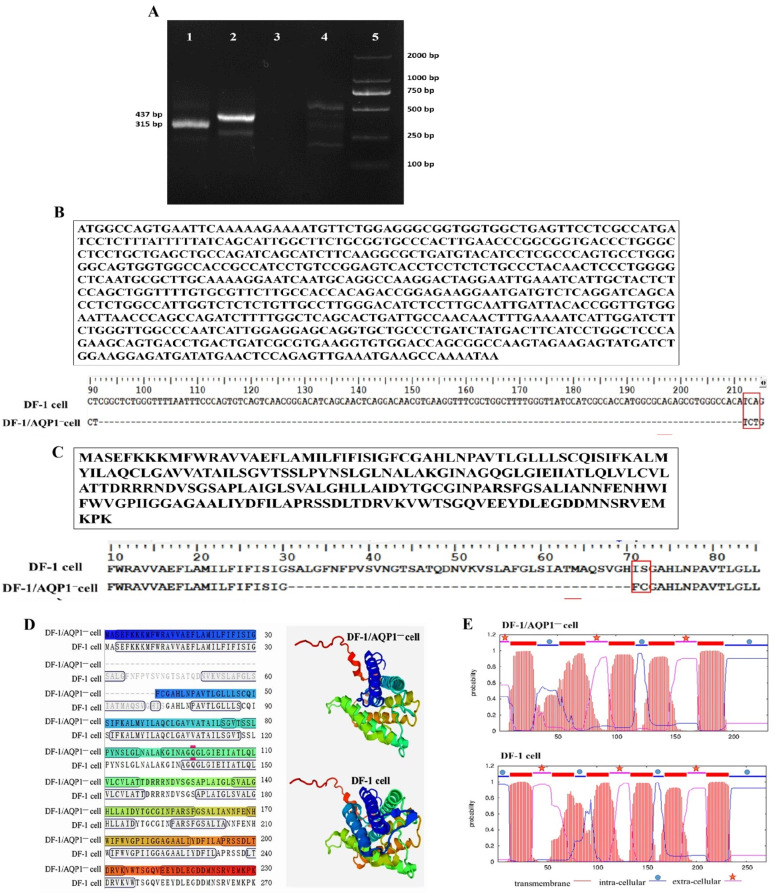
Identification of DF-1/AQP1^−^ cells. (**A**). RT-PCR analysis—12345, respectively, stands for (1. DF-1/AQP1^−^ cells; 2. DF-1 cells; 3. The blank control; 4. The complete cell culture medium; 5. The marker). (**B**). Sequencing of the AQP1 mutant gene in the DF-1/AQP1^−^ cell clone and comparative alignment with the wild-type AQP1 sequence. (**C**). Predicted amino acid sequence of the AQP1 mutant protein in DF-1/AQP1^−^ cells and comparison with the wild-type AQP1 protein. (**D**). Three-dimensional structural comparison between the AQP1 mutant protein in DF-1/AQP1^−^ cells and the wild-type AQP1 protein in DF-1 cells. (**E**). Comparison of the predicted transmembrane domain architecture of AQP1 in DF-1/AQP1^−^ cells and wild-type DF-1 cells.

**Figure 2 vaccines-14-00052-f002:**
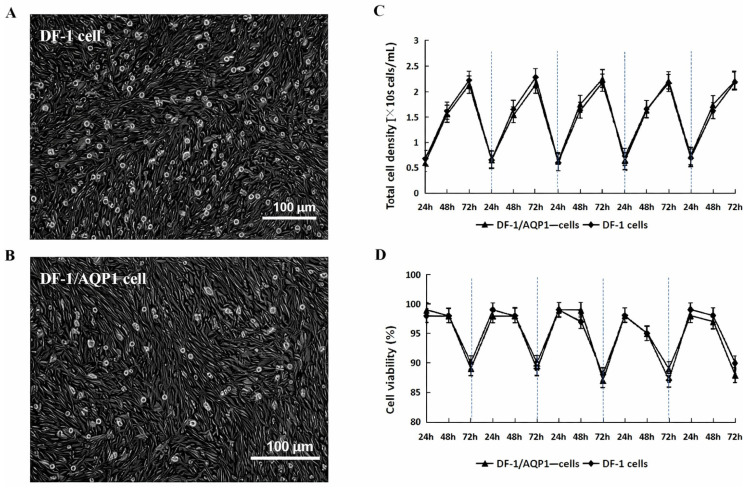
Growth characteristics of DF-1 and DF-1/AQP1^−^ cells in T-flask culture. (**A**) Morphology of wild-type DF-1 cells grown as a monolayer (original magnification, ×100). (**B**) Morphology of DF-1/AQP1^−^ cells grown as a monolayer (original magnification, ×100). (**C**) Growth kinetics of DF-1 and DF-1/AQP1^−^ cells. (**D**) Cell viability of DF-1 and DF-1/AQP1^−^ cells during continuous passage. A total of 1 × 10^6^ DF-1/AQP1^−^ cells or DF-1 cells were seeded in 25 cm^2^ T-flasks, and their growth kinetics and viability were evaluated during serial passaging.

**Figure 3 vaccines-14-00052-f003:**
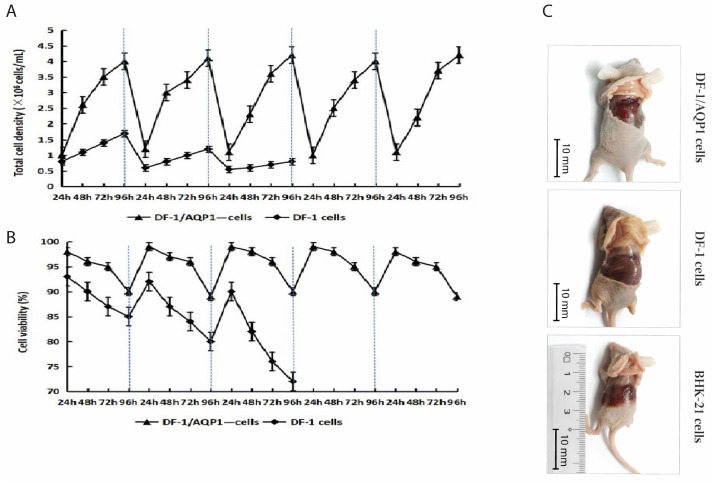
Growth characteristics of DF-1/AQP1^−^ cells in suspension culture and tumorigenicity assessment. DF-1/AQP1^−^ and wild-type DF-1 cells were seeded at 1.5 × 10^7^ cells in 125 mL vented shake flasks containing 30 mL DMEM supplemented with 5% (*v*/*v*) fetal bovine serum. (**A**). Growth kinetics of DF-1/AQP1^−^ and DF-1 cells in suspension culture. (**B**). Cell viability of DF-1/AQP1^−^ and DF-1 cells during suspension culture. (**C**). Tumorigenicity assessment of DF-1/AQP1^−^ cells after subcutaneous injection into the shoulder blade area of nude mice.

**Table 1 vaccines-14-00052-t001:** List of sgRNA and primer sequences.

Name	Forward/Reverse Sequence (5′–3′)
sgRNA AQP1-g1	ATCAGCATTGGCTCGGCTCTGGG
sgRNA AQP1-g2	GTGGCCCACGCTCTGCGCCATGG
AQP1-g1 reverse transcription primer	TAATACGACTCACTATAGGGATCAGCATTGGCTCGGCTCTGTTCAGAGCTATGCTGGA
AQP1-g2 reverse transcription primer	TAATACGACTCACTATAGGGGTGGCCCACGCTCTGCGCCAGTTCAGAGCTATGCTGGA
RT-PCR identification primer	AQP1-Forward GGGCAGAGAAAGAGGAGAGAT AQP1-Reverse AGTAGAGGGGGACAGCAAAGT
Sequence analysis primer	AQP1-Forward GAGGGCAGAGAAAGAGGAGAG AQP1-Reverse CTTCTCCTTCTTTATTTTGGCTTC

sgRNA: single guide RNA, RT-PCR: Reverse Transcription-Polymerase Chain Reaction, AQP1.

**Table 2 vaccines-14-00052-t002:** Virus titers in different cell substrates.

Substrate	Viral Titer	
	Log TCID_50_/mL	Log EID_50_/mL
DF-1/AQP1^−^ cells suspension	9.0 ± 0.12	8.63 ± 0.13
DF-1/AQP1^−^ cells monolayer	7.21 ± 0.07	7.14 ± 0.13
DF-1 cells monolayer	7.38 ± 0.13	7.04 ± 0.07

## Data Availability

Data will be made available on request.

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
