# Peer review of "Engineering AQP1-Deficient DF-1 Suspension Cells for High-Yield IBDV Production and Vaccine Scale-Up"

_vaccines, 2025, doi:10.3390/vaccines14010052_

Round 1

Reviewer 1 Report

Comments and Suggestions for Authors

This manuscript presents a significant technological advance in the manufacturing of vaccines against Infectious Bursal Disease Virus. IBD is a severe immunosuppressive poultry disease, and its control relies heavily on vaccination. Vaccine production typically utilizes DF-1 chicken embryo fibroblast cells grown in adherent culture (e.g., roller bottles or multi-layer flasks). This method is complex, labor-intensive, and difficult to scale up cost-effectively for mass production.

The authors directly address this manufacturing barrier by developing an engineered DF-1 cell line that can grow in serum-free suspension culture. This transition to suspension culture is the gold standard for high-volume vaccine production using bioreactors, as it offers superior scalability and lower operational costs. The central innovation is the targeted engineering of the DF-1 cells to be Aquaporin 1 (AQP1) deficient. AQP1 is a water channel protein. Deleting this protein likely enhances the cell's robustness by optimizing its ability to regulate cell volume and cope with osmotic stress when transitioning from an adherent, serum-rich environment to a serum-free suspension medium.

Based on the observations and recommendations in this correspondence, here are key questions that could be posed to the authors:

  1. Was the protective efficacy of the IBDV vaccine produced in the AQP1-deficient suspension cells tested in chickens via a virulent IBDV challenge? Specifically, how do the serum neutralizing antibody titers and Bursal Lesion Scores in challenged chickens compare between the vaccine produced in the engineered cells versus the vaccine produced in the parental wild-type adherent cells?
  2. How many continuous passages was the AQP1-deficient cell line cultured in the serum-free suspension medium? Was the high IBDV yield and the AQP1-deficient genotype (e.g., via PCR or sequencing) maintained stably throughout the maximum tested passage number?
  3. Besides the achievement of a higher maximum viable cell density, did the researchers observe any changes in the IBDV replication kinetics or virus release efficiency (e.g., burst size per cell) in the AQP1-deficient cells compared to the parental cells? Does the AQP1 deletion directly influence viral processes beyond just cell growth?
  4. Were the high-yield production results validated in a pilot-scale bioreactor (e.g., a minimum of a 3L stirred-tank system) to confirm industrial feasibility? What is the estimated reduction in the cost of goods sold (COGS) per vaccine dose afforded by this AQP1-deficient suspension system compared to a traditional adherent production system?

Author Response

Reviewer -1

suggestions

This manuscript presents a significant technological advance in the manufacturing of vaccines against Infectious Bursal Disease Virus. IBD is a severe immunosuppressive poultry disease, and its control relies heavily on vaccination. Vaccine production typically utilizes DF-1 chicken embryo fibroblast cells grown in adherent culture (e.g., roller bottles or multi-layer flasks). This method is complex, labor-intensive, and difficult to scale up cost-effectively for mass production.

The authors directly address this manufacturing barrier by developing an engineered DF-1 cell line that can grow in serum-free suspension culture. This transition to suspension culture is the gold standard for high-volume vaccine production using bioreactors, as it offers superior scalability and lower operational costs. The central innovation is the targeted engineering of the DF-1 cells to be Aquaporin 1 (AQP1) deficient. AQP1 is a water channel protein. Deleting this protein likely enhances the cell's robustness by optimizing its ability to regulate cell volume and cope with osmotic stress when transitioning from an adherent, serum-rich environment to a serum-free suspension medium.

Response: We sincerely thank the reviewer for the careful evaluation of our work and the positive recognition of our experimental approach and methodology. We appreciate these constructive comments and have addressed each of the concerns in detail below.

Point 1: Based on the observations and recommendations in this correspondence, here are key questions that could be posed to the authors: Was the protective efficacy of the IBDV vaccine produced in the AQP1-deficient suspension cells tested in chickens via a virulent IBDV challenge? Specifically, how do the serum neutralizing antibody titers and Bursal Lesion Scores in challenged chickens compare between the vaccine produced in the engineered cells versus the vaccine produced in the parental wild-type adherent cells?

Response:  According to the requirements of the Pharmacopoeia of Veterinary Drugs of the People's Republic of China, the efficacy test of IBD live vaccines can be conducted using the test method with chicken embryo (The SPF chicken embryos aged 10-12 days are inoculated with the IBDV virus, and then the EID50 was evaluated) or test method with the chicken (The 20 days after vaccination of SPF chickens aged 14-28 days, IBDV strong virus is used for attack to evaluate the protective efficiency of the vaccine). But unfortunately,we selected the first method to evaluate the efficacy of IBDV in our study.  But we have incorporated this valuable suggestion into our subsequent research plan regarding the IBDV vaccine to further improve the research system.

Point2: How many continuous passages was the AQP1-deficient cell line cultured in the serum-free suspension medium? Was the high IBDV yield and the AQP1-deficient genotype (e.g., via PCR or sequencing) maintained stably throughout the maximum tested passage number?

Response: According to the vaccine production with cells, it is stipulated that the cell generations of the basic cell bank should be limited to within five generations. And the cell generations of the working cell bank are limited to within 10 generations. And then, the cell generations for production should be limited to within 20 generations. In the study, we demonstrated that the AQP1 mutant gene in DF-1/AQP1⁻ cells remained stable over 20 consecutive passages, and without evidence of revertant mutations (In the section 3.5). On the other hand, the cells used for IBDV cultivation were generally maintained at around 15 generations during the experiment. However, we did not deliberately use the 20th generation cells to cultivate IBDV and to evaluate its viral titer. But based on my understanding of DF-1/AQP1⁻ cells and my experimental experience, I believe that using the 20th generation DF-1/AQP1⁻ cells to culture IBDV would also obtain the high yield IBDV.

Point 3: Besides the achievement of a higher maximum viable cell density, did the researchers observe any changes in the IBDV replication kinetics or virus release efficiency (e.g., burst size per cell) in the AQP1-deficient cells compared to the parental cells? Does the AQP1 deletion directly influence viral processes beyond just cell growth?

Response: In the section 3.3.1, we have displayed that “In monolayer culture, the DF-1/AQP1⁻ cells and wild-type DF-1 cells exhibited the comparable growth kinetics and viability (P > 0.05)”. 

In the section 3.6, we have already made the description. When using the DF-1/AQP1⁻ cells and wild-type DF-1 cells to culture IBDV in monolayer culture, there was no significant difference in the time of virus harvest and the virus titer. The result showed that the mutant of AQP1 gene did not alter the cell’s susceptibility and the viral processes. But, in suspension culture, the time of virus harvest reduced to 36-48h and the virus titer obviously increased.

In the discussion, we also made the discussion about the viral harvest time and viral titer during suspension culture. This improvement of viral titer is likely due to the higher cell density achievable in suspension and the more uniform distribution of virus throughout the medium facilitated by stirring or shaking. Consequently, culture duration is reduced, and viral production is markedly enhanced.

Point 4. Were the high-yield production results validated in a pilot-scale bioreactor (e.g., a minimum of a 3L stirred-tank system) to confirm industrial feasibility? What is the estimated reduction in the cost of goods sold (COGS) per vaccine dose afforded by this AQP1-deficient suspension system compared to a traditional adherent production system?

Response: In this study, we only produced IBDV with shake flasks, without conducting pilot-scale production with bioreactors. But in the subsequent research on the production of IBDV vaccine, we will culture the DF-1/AQP1⁻ cells and IBDV in the bioreactor.

In the discussion, based on the current production situation of the IBDV vaccine, we compared the culture value of each milliliter of the virus solution between the traditional egg-based production and suspension culture. The value of each milliliter of IBDV in suspension culture decreases by 7.28 times to 10.91 times compared with the traditional egg-based vaccine production.

Reviewer 2 Report

Comments and Suggestions for Authors

In this study (Engineering AQP1-Deficit DF-1 suspension cells for high-yield IBDV production and vaccine scale-up), a DF-1 cell with AQP1 gene deletion (DF-1/AQP1) was constructed by CRISPR/Cas9 technology, which successfully realized its suspension culture without adaptation, and significantly increased the output of infectious bursal disease virus (IBDV). The research design is reasonable, the experimental system is complete and the data is detailed, which provides a potential cell platform for the large-scale production of poultry vaccines. The writing of the paper is clear and complete, which has some application value and scientific significance. Hower, the manuscript still has the following problems that need to be improved:

  1. Figure 1 is fuzzy as a whole, especially Figure 1A is of poor quality, which makes it unclear. In addition, what 1234 stands for is not clearly explained.

2.Figs. 2A and 2B are not to scale.

3.Figure 3C, the background is inconsistent, the ruler is not suitable, and it is not clear.

  1. p of statistics has to write italic, and there are many problems like this
  2. The wild-type DF-1 cells cultured in suspension in Figure 3 cannot be maintained in the fourth generation, is it related to apoptosis or aggregation? It is suggested to supplement the relevant discussion.

6.Does the strategy of glucose supplementation in suspension culture in section 6 2.9.2 affect virus replication?

  1. To supplement the gap and demand of suspended cell platform in poultry vaccine production at present, and further highlight the significance of this study.
  2. Compare the advantages and limitations of this system with other suspended cell platforms (such as BHK-21 and MDCK) in virus production and culture cost.
  3. Further explore the molecular mechanism of suspension growth caused by AQP1 deletion, especially the relationship with cell adhesion pathways (such as integrin and cadherin).

Author Response

Reviewer -2

suggestions

In this study (Engineering AQP1-Deficit DF-1 suspension cells for high-yield IBDV production and vaccine scale-up), a DF-1 cell with AQP1 gene deletion (DF-1/AQP1) was constructed by CRISPR/Cas9 technology, which successfully realized its suspension culture without adaptation, and significantly increased the output of infectious bursal disease virus (IBDV). The research design is reasonable, the experimental system is complete and the data is detailed, which provides a potential cell platform for the large-scale production of poultry vaccines. The writing of the paper is clear and complete, which has some application value and scientific significance. Hower, the manuscript still has the following problems that need to be improved:

Response: We sincerely thank the reviewer for the careful evaluation of our work and the positive recognition of our experimental approach and methodology. We appreciate these constructive comments and have addressed each of the concerns in detail below.

Point 1: Figure 1 is fuzzy, especially Figure 1A is of poor quality, which makes it unclear. In addition, what 1234 stands for is not clearly explained.

Response: Thank you for your valuable comment. According to your comments, we have improved the quality of Figure 1A.

Point 2: Figs. 2A and 2B are not to scale.

Response: We really appreciate your advice. In the Figure 2A and 2B, we have increased the contrast of pictures, and we have also inserted the ruler bar.

Point 3: Figure 3C, the background is inconsistent, the ruler is not suitable, and it is not clear.

Response: Thank you so much for your input. In the Figure 3C, the background of the three pictures has been modified, and they are all the same now.

Point 4: p of statistics has to write italic, and there are many problems like this

Response: Thank you so much for your comment. Accordingly, In the results section, all the ‘’p’’ has changed into italics.

Point 5: The wild-type DF-1 cells cultured in suspension in Figure 3 cannot be maintained in the fourth generation, is it related to apoptosis or aggregation? It is suggested to supplement the relevant discussion.

Response: Thanks so much. In the section 3.3.2 and discussion section, we have displayed of the cell states, as well as the description of the death status of wild-type DF-1 cells during suspension culture.

Point 6: Does the strategy of glucose supplementation in suspension culture in section 6 2.9.2 affect virus replication?

Response: Thank you for your comment. During the monolayer culture process of wild-type DF-1 cells and DF-1/AQP1⁻cells, the DMEM medium used was a high-glucose medium. Therefore, in the suspension culture process, we added glucose into the medium. On the other hand, the consumption of glucose can, to a certain extent, be used to evaluate the growth and metabolic level of cells.

Point 7: To supplement the gap and demand of suspended cell platform in poultry vaccine production at present and further highlight the significance of this study.

Response: We are very grateful for your insight. In the Introduction, we supplement the gap and demand of suspended cell platform in poultry vaccine production at present, and that highlight the significance of this study.

Point 8: Compare the advantages and limitations of this system with other suspended cell platforms (such as BHK-21 and MDCK) in virus production and culture cost.

Response: In the discussion, based on the current production situation of the IBDV vaccine, we compared the culture value of each milliliter of the virus solution between the traditional egg-based production and suspension culture. Maybe, the other suspended cell platforms (such as BHK-21 and MDCK) in virus production are better, but these cells are not suitable for IBDV.

Point 9: Further explore the molecular mechanism of suspension growth caused by AQP1 deletion, especially the relationship with cell adhesion pathways (such as integrin and cadherin).

Response: In the discussion, we have discussed the possible mechanism of how aquaporin mutation may contribute to cell’s suspension growth. It has been reported that the mutant of AQP1 can obviously inhibit protein expression of adhesion molecules (VCAM-1 and ICAM-1). Therefore, the adhesiveness of the cells significantly decrease when the AQP1 gene is deleted or mutated.  On the other hand, our main research objective is to construct a DF-1 suspension cell line by knocking out the AQP1 gene, thereby promoting the proliferation of avian viruses and providing a reference for the application of bioreactors in vaccine production. Regarding the supplementary experiment about the mechanistic link between AQP1 mutant and loss of adhesion, due to the constraints of the existing fund of the project, it is temporarily impossible to complete the relevant research work in the short term. We have incorporated this valuable suggestion into our subsequent research plan and intend to conduct this part of the experiment specifically in another subsequent article to further improve the research system.

Reviewer 3 Report

Comments and Suggestions for Authors

The manuscript by Bingmei Dong et al., is devoted the development of a stable DF-1 suspension cell line via AQP1 disruption and evaluation its potential for enhanced infectious bursal disease virus (IBDV) production. The manuscript sounds scientific, the title and abstract correspond to the content, and will be of interest to researchers in this field. However, I believe that in the current state the manuscript needs revision.

  1. Fig. 2A and 2B: On these images, it is necessary to increase the contrast, because cells are hard to see. Besides, there is no ruler bar in the picture, please insert.
  2. How can the authors explain why the titer in DF1 and AQP-KO in the monolayer is the same, does this mean that the mutation does not affect the viral titer (Table 2). If this is the case, then I suggest the authors should insert this as a result.
  3. I believe, the authors do not discuss their results enough. In my opinion, the authors should expand the Discussion section by comparing their approach to increasing virus production with similar studies on increasing virus production and saying why their approach is better or worse than other approaches. In particular, using of small beads (microcarriers) to increase the surface for attachment of DF1 cells, allowing them to grow in suspension, enabling large-scale growth for viruses. Another approach is genetic modifications in DF1 cells, which promote greater virus production than unmodified cells.

Author Response

suggestions

The manuscript by Bingmei Dong et al., is devoted the development of a stable DF-1 suspension cell line via AQP1 disruption and evaluation its potential for enhanced infectious bursal disease virus (IBDV) production. The manuscript sounds scientific; the title and abstract correspond to the content and will be of interest to researchers in this field. However, I believe that in the current state the manuscript needs revision.

Response: We sincerely thank the reviewer for the careful evaluation of our work and the positive recognition of our experimental approach and methodology. We appreciate these constructive comments and have addressed each of the concerns in detail below.

Point 1- Fig. 2A and 2B: On these images, it is necessary to increase the contrast, because cells are hard to see. Additionally, there is no ruler bar in the picture; please insert it.

 Response: Thank you so much for your comments on our figures.  Accordingly, in Figures 2A and 2B, we have increased the contrast of pictures and inserted the ruler bar.

Point 2: How can the authors explain why the titer in DF1 and AQP-KO in the monolayer is the same, does this mean that the mutation does not affect the viral titer (Table 2). If this is the case, then I suggest the authors should insert this as a result.

 Response: Thank you so much for the valuable point your raised.  In our study, the mutant of AQP1 gene obviously decreased the adhesiveness of DF-1 cells but does not affect the susceptibility of DF-1 cells to IBDV. So, the amount of virus is about the same in both wild-type DF-1 and DF-1/AQP1⁻ However, DF-1/AQP1⁻cells in suspension exhibited significantly higher TCID50 and EID50 values compared with monolayer culture of DF-1 and DF-1/AQP1⁻ cells.  It may be that the suspension culture significantly increased the number of DF-1/AQP1⁻ cells per unit volume, thereby enhancing the viral titer.  In section 3.6, we displayed the result.

Point 3: I believe the authors do not discuss their results enough. In my opinion, the authors should expand the Discussion section by comparing their approach to increasing virus production with similar studies on increasing virus production and saying why their approach is better or worse than other approaches. In particular, small beads (microcarriers) increase the surface for DF1 cell attachment, allowing them to grow in suspension and enabling large-scale virus production. Another approach is genetic modifications in DF1 cells, which promote greater virus production than unmodified cells.

Response: We appreciate this valuable suggestion, which has helped us further strengthen the manuscript. Following your valuable comments, we have tried to compare the three types of large-scale cell production technologies and demonstrated the advantages of using CRISPR technology, which mutates cells and makes them suitable for suspension culture techniques.

Round 2

Reviewer 1 Report

Comments and Suggestions for Authors

This manuscript presents a significant technological advance in the manufacturing of vaccines against Infectious Bursal Disease Virus. IBD is a severe immunosuppressive poultry disease, and its control relies heavily on vaccination. Vaccine production typically utilizes DF-1 chicken embryo fibroblast cells grown in adherent culture (e.g., roller bottles or multi-layer flasks). This method is complex, labor-intensive, and difficult to scale up cost-effectively for mass production.

The authors directly address this manufacturing barrier by developing an engineered DF-1 cell line that can grow in serum-free suspension culture. This transition to suspension culture is the gold standard for high-volume vaccine production using bioreactors, as it offers superior scalability and lower operational costs. The central innovation is the targeted engineering of the DF-1 cells to be Aquaporin 1 (AQP1) deficient. AQP1 is a water channel protein. Deleting this protein likely enhances the cell's robustness by optimizing its ability to regulate cell volume and cope with osmotic stress when transitioning from an adherent, serum-rich environment to a serum-free suspension medium.

Based on the observations and recommendations in this correspondence, here are key questions that could be posed to the authors:

  1. Was the protective efficacy of the IBDV vaccine produced in the AQP1-deficient suspension cells tested in chickens via a virulent IBDV challenge? Specifically, how do the serum neutralizing antibody titers and Bursal Lesion Scores in challenged chickens compare between the vaccine produced in the engineered cells versus the vaccine produced in the parental wild-type adherent cells?
  2. How many continuous passages was the AQP1-deficient cell line cultured in the serum-free suspension medium? Was the high IBDV yield and the AQP1-deficient genotype (e.g., via PCR or sequencing) maintained stably throughout the maximum tested passage number?
  3. Besides the achievement of a higher maximum viable cell density, did the researchers observe any changes in the IBDV replication kinetics or virus release efficiency (e.g., burst size per cell) in the AQP1-deficient cells compared to the parental cells? Does the AQP1 deletion directly influence viral processes beyond just cell growth?
  4. Were the high-yield production results validated in a pilot-scale bioreactor (e.g., a minimum of a 3L stirred-tank system) to confirm industrial feasibility? What is the estimated reduction in the cost of goods sold (COGS) per vaccine dose afforded by this AQP1-deficient suspension system compared to a traditional adherent production system?

Author Response

Reviewer -1

suggestions

This manuscript presents a significant technological advance in the manufacturing of vaccines against Infectious Bursal Disease Virus. IBD is a severe immunosuppressive poultry disease, and its control relies heavily on vaccination. Vaccine production typically utilizes DF-1 chicken embryo fibroblast cells grown in adherent culture (e.g., roller bottles or multi-layer flasks). This method is complex, labor-intensive, and difficult to scale up cost-effectively for mass production.

The authors directly address this manufacturing barrier by developing an engineered DF-1 cell line that can grow in serum-free suspension culture. This transition to suspension culture is the gold standard for high-volume vaccine production using bioreactors, as it offers superior scalability and lower operational costs. The central innovation is the targeted engineering of the DF-1 cells to be Aquaporin 1 (AQP1) deficient. AQP1 is a water channel protein. Deleting this protein likely enhances the cell's robustness by optimizing its ability to regulate cell volume and cope with osmotic stress when transitioning from an adherent, serum-rich environment to a serum-free suspension medium.

Response: We sincerely thank the reviewer for the careful evaluation of our work and the positive recognition of our experimental approach and methodology. We appreciate these constructive comments and have addressed each of the concerns in detail below.

Point 1: Based on the observations and recommendations in this correspondence, here are key questions that could be posed to the authors: Was the protective efficacy of the IBDV vaccine produced in the AQP1-deficient suspension cells tested in chickens via a virulent IBDV challenge? Specifically, how do the serum neutralizing antibody titers and bursal lesion scores in challenged chickens compare between the vaccine produced in the engineered cells versus the vaccine produced in the parental wild-type adherent cells?

Response:  According to the requirements of the Pharmacopoeia of Veterinary Drugs of the People's Republic of China, the efficacy test of IBD live vaccines can be conducted using the test method with chicken embryo (The SPF chicken embryos aged 10-12 days are inoculated with the IBDV virus, and then the EID50 was evaluated) or test method with the chicken (The 20 days after vaccination of SPF chickens aged 14-28 days, IBDV strong virus is used for attack to evaluate the protective efficiency of the vaccine). But unfortunately,we selected the first method to evaluate the efficacy of IBDV in our study.  But we have incorporated this valuable suggestion into our subsequent research plan regarding the IBDV vaccine to further improve the research system.

Point2: How many continuous passages was the AQP1-deficient cell line cultured in the serum-free suspension medium? Was the high IBDV yield and the AQP1-deficient genotype (e.g., via PCR or sequencing) maintained stably throughout the maximum tested passage number?

Response: According to the vaccine production with cells, it is stipulated that the cell generations of the basic cell bank should be limited to within five generations. And the cell generations of the working cell bank are limited to within 10 generations. And then, the cell generations for production should be limited to within 20 generations. In the study, we demonstrated that the AQP1 mutant gene in DF-1/AQP1⁻ cells remained stable over 20 consecutive passages, and without evidence of revertant mutations (In the section 3.5). On the other hand, the cells used for IBDV cultivation were generally maintained at around 15 generations during the experiment. However, we did not deliberately use the 20th generation cells to cultivate IBDV and to evaluate its viral titer. But based on my understanding of DF-1/AQP1⁻ cells and my experimental experience, I believe that using the 20th generation DF-1/AQP1⁻ cells to culture IBDV would also obtain the high yield IBDV.

Point 3: Besides the achievement of a higher maximum viable cell density, did the researchers observe any changes in the IBDV replication kinetics or virus release efficiency (e.g., burst size per cell) in the AQP1-deficient cells compared to the parental cells? Does the AQP1 deletion directly influence viral processes beyond just cell growth?

Response: In the section 3.3.1, we have displayed that “In monolayer culture, the DF-1/AQP1⁻ cells and wild-type DF-1 cells exhibited the comparable growth kinetics and viability (P > 0.05)”. 

In the section 3.6, we have already made the description. When using the DF-1/AQP1⁻ cells and wild-type DF-1 cells to culture IBDV in monolayer culture, there was no significant difference in the time of virus harvest and the virus titer. The result showed that the mutant of AQP1 gene did not alter the cell’s susceptibility and the viral processes. But, in suspension culture, the time of virus harvest reduced to 36-48h and the virus titer obviously increased.

In the discussion, we also made the discussion about the viral harvest time and viral titer during suspension culture. This improvement of viral titer is likely due to the higher cell density achievable in suspension and the more uniform distribution of virus throughout the medium facilitated by stirring or shaking. Consequently, culture duration is reduced, and viral production is markedly enhanced.

Point 4. Were the high-yield production results validated in a pilot-scale bioreactor (e.g., a minimum of a 3L stirred-tank system) to confirm industrial feasibility? What is the estimated reduction in the cost of goods sold (COGS) per vaccine dose afforded by this AQP1-deficient suspension system compared to a traditional adherent production system?

Response: In this study, we only produced IBDV with shake flasks, without conducting pilot-scale production with bioreactors. But in the subsequent research on the production of IBDV vaccine, we will culture the DF-1/AQP1⁻ cells and IBDV in the bioreactor.

In the discussion, based on the current production situation of the IBDV vaccine, we compared the culture value of each milliliter of the virus solution between the traditional egg-based production and suspension culture. The value of each milliliter of IBDV in suspension culture decreases by 7.28 times to 10.91 times compared with the traditional egg-based vaccine production.

Reviewer 2 Report

Comments and Suggestions for Authors

I have no other comments

Author Response

Response: Thank you for your comment.

Reviewer 3 Report

Comments and Suggestions for Authors

The authors have provided responses to all my comments, therefore I do not have any further suggestions.

Author Response

Response: We appreciate this valuable suggestion.